# NCDL: A Framework for Deep Learning on non-Cartesian Lattices

**Joshua J. Horacsek**
Department of Computer Science
University of Calgary
Calgary, Alberta
j.horacsek@ncdl.ai

**Usman R. Alim**
Department of Computer Science
University of Calgary
Calgary, Alberta
ualim@ucalgary.ca

## Abstract

The use of non-Cartesian grids is a niche but important topic in sub-fields of the numerical sciences, such as simulation and scientific visualization. However, non-Cartesian approaches are virtually unexplored in machine learning. This is likely due to the difficulties in the representation of data on non-Cartesian domains and the lack of support for standard machine learning operations on non-Cartesian data. This paper proposes a new data structure called the *lattice tensor* which generalizes traditional tensor spatio-temporal operations to lattice tensors, enabling the use of standard machine learning algorithms on non-Cartesian data. We introduce a software library that implements this new data structure and demonstrate its effectiveness on various problems. Our method provides a general framework for machine learning on non-Cartesian domains, addressing the challenges mentioned above and filling a gap in the current literature.

## 1 Introduction

Machine learning heavily relies on *tensors* to represent multi-dimensional data. However, tensors are inherently Cartesian, and representing data solely on Cartesian grids can be restrictive. Certain data may be more naturally suited to alternative grid structures. For example, raw image data from most imaging sensors is not Cartesian; a Bayer filter represents blue and red data with Cartesian structure, but green data has quincunx (checkerboard) structure. There is also abundant literature showing that hexagonal grids are superior to Cartesian grids when representing isotropically band-limited natural images [4, 34, 32, 38, 3]. Representing data in these domains using traditional Cartesian tensors can be inefficient and lead to inaccuracies depending on the distribution the data belongs to. For the distribution of natural images, which have relatively isotropic behaviour in the Fourier domain, it is well known that using alternative grids yields better image representations [38].

Another reason to use exotic grids is that alternative structures can have significantly different approximation capabilities. Again, turning to the hexagonal lattice as an example, which is well known as the optimal sampling lattice in 2D [4], if one samples a 2D signal with a number of samples, a hexagonal sampling reduces the memory consumption by approximately 14% [32]. This follows from a simple argument in the Fourier domain; sampling a band-limited signal periodizes that function in the Fourier domain. If one's sampling pattern is hexagonal (the optimal circle packing pattern) then less information is lost when the signal is hexagonally periodized. This effect compounds as dimensionality increases; for example the body-centered-cubic (BCC) lattice is 30% more efficient than the Cartesian lattice (for isotropically band-limited functions)[1].

However, though these grid structures show promise in other areas of numerical sciences, they are relatively unexplored in the context of machine learning. There is no user-friendly, general, and

---

[1]The face-centered-cubic (FCC) lattice is also attractive, as it minimizes aliasing.

37th Conference on Neural Information Processing Systems (NeurIPS 2023).

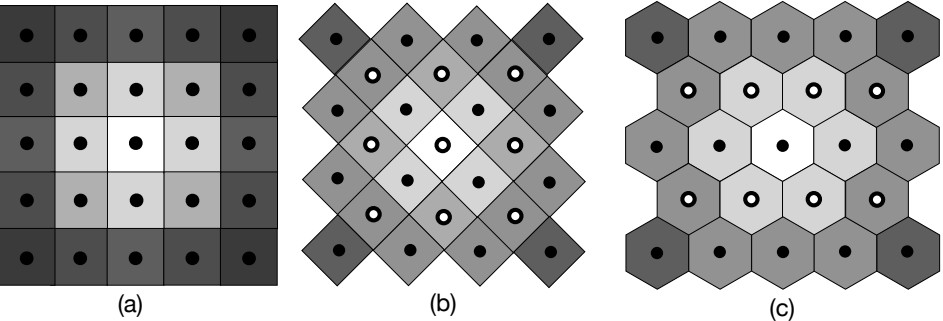

Figure 1: Three lattices used as examples throughout this paper; a) is the Cartesian lattice, b) is the quincunx lattice and c) is the hexagonal lattice. The cells of each lattice have been shaded according to their distance to the origin. We ignore the cell structure of the lattice, and treat a lattice structure as a simple point structure. The quincunx and hexagonal lattices are formed by interleaving two Cartesian grids; this is shown with the white and black filled dots.

efficient way to represent data on non-Cartesian grids within the context of machine learning (or in general). To address this gap, we introduce a new data-structure called a *lattice tensor* that we use to represent data on any non-Cartesian lattice structure. This structure changes the inherent representation of the data, as such it requires many common operations to be rewritten in terms of lattice tensors. For the most part, this is straightforward. For example, element-wise operations are trivial to extend. Convolution is more involved, but we still may leverage existing fast implementations. We generalize many of the traditional tensor spatio-temporal operations to lattice tensors, enabling the use of these operations on non-Cartesian domains.

Ultimately, we present a software library called *Non-Cartesian Deep Learning* (NCDL) which is an open source, concrete implementation of the lattice tensor container and the associated spatio-temporal operations defined over lattice tensors. NCDL library is implemented on top of PyTorch [31], and is designed to be relatively easily integrated into existing PyTorch code bases.

This work is the first general data processing library for non-Cartesian data (with the added benefit of being differentiable). In short, our contributions are as follows:

1. We introduce the concept of a lattice tensor, a data-structure for the representation of data on non-Cartesian lattice structures.

2. We generalize the traditional spatio-temporal operations such as convolution and pooling to lattice tensors.

3. We provide an open source software library that implements our methodology.

4. We show a small set of cases in which the non-Cartesian ideology produces a performance benefit in the broader context of machine learning.

The remainder of the paper is organized as follows. In Section 2, we review related work on non-Cartesian grids both in and out of the context of machine learning. In Section 3, we formalize the key operations and concepts that we use in NCDL. In Section 4, we validate the performance of our implementation for convolution against a specialized implementation for hexagonal lattices, and explore non-Cartesian auto-encoding networks. Finally, in Section 5, we conclude and discuss future work. Our implementation is available at `https://www.ncdl.ai`.

## 2   Related Work

Convolution is a fundamental operation in most machine vision pipelines, and can be traced back to the early 1980s [12]. Convolutional networks gained popularity owing to the fact that convolutions exploit the locality of data while avoiding the curse of dimensionality (in parameter count). The recent surge of success in convolutional neural networks (CNNs) is in large part to hardware advances in the early 2010s. General purpose GPU compute power facilitated new benchmark performance in image classification [25, 1]. Since then, various forms of CNNs have been proposed with various design differences: early efforts, like VGG and ResNet, focused on building deeper convolutional

networks [14, 29] while trying to avoid the vanishing gradient problem; while others attempted to create more computationally/parameter efficient networks [18, 33, 36, 37]. Other works have focused on changing the mechanics of the convolution layer [44, 35, 18]. However, none of these have pushed into the realm of non-Cartesian data.

There has been limited work on the use of non-Cartesian grids in machine learning. The space efficiency of the hexagonal lattice has been attractive, so it has garnered a small amount of attention. *NeuroHex* was the first to explore hexagonal convolutions in a modern CNN setting; their domain is the game of *Hex* [43]. The game of Hex is restricted to a parallelpiped shaped region; to support hexagonal convolution, the authors simply transform the problem domain and restrict convolutions to the hexagonal network by zeroing out filter elements. *HexaConv* explores hexagonal convolution in the context of image classification, and shows a surprising improvement over traditional Cartesian convolution [15]. They resample square Cartesian images on to the hexagonal lattice in an axial coordinate system (i.e. the same parallelpiped as [43], but zero padded). A similar work proposed a nearly identical structure on the isocsahedron-based spherical hexagonal grid. These works all suffer from the same problem: wasted space in their representation. There are currently only two works that do not waste space in this manner. The first is *HexCNN* [47], which stores data in a column format effectively linearizing the hexagonal domain column by column; convolutions then operate on this domain. The second and most similar to our work is HexagDLy, which splits the convolution into multiple convolutions over different shifts of the input domain [34]. Our work is more general than all of these works; our primary contribution is the lattice tensor, a simple, waste-free container for data on arbitrary lattices. Figure 2 shows alternative representations from other frameworks, where they waste memory, and how NCDL addresses this.

This generalization forces us to think more broadly about operations on lattice tensors. Take down sampling for example; traditionally, down sampling is uniform in each dimension (usually dyadic, i.e. a factor of 2 in each dimension). However, this is a limitation only imposed by the restriction to Cartesian grids. There are other possible decimation strategies when discarding samples on the Cartesian lattice, one may discard samples from a checkerboard (i.e. quincunx) pattern, thereby producing another quincunx lattice. This type of downsampling is present in non-dyadic downsampling multi-channel wavelet filterbanks [24], but has not been explored in the context of machine learning. In fact, higher dimensions exhibit more interesting cascading structures between grids [19]. This cascading structure is something we explicitly support in NCDL.

The benefit of non-Cartesian grids extends to higher dimensions. The optimal packing lattice in 3D is the body-centered cubic (BCC) lattice, and this is relatively well studied in the context of scientific visualization [32, 4]. However, most research has focused on finding high performing interpolants in these spaces [9, 22, 11, 21, 7, 10, 6], or on extending numerical techniques to exploit the inherent benefit of these approximation spaces. In this sense, we also contribute in this area, by providing an interface to implement these ideas (one is not required to use the differentiablity provided by NCDL).

Another related subfield of machine learning is *graph learning* [42]. Examples of graph learning problems include document classification, where documents are the set of nodes $V$, and are related to other nodes/documents through the edge set $E$; this set may represent academic citiations, for example [28]. Since non-Cartesian lattices change the inherent structure of the underlying representation of the grid, it is natural to draw connections between non-Cartesian methods and graph based approaches to operations such as convolution and pooling [13, 26, 27]. However, upon closer inspection, these connections break down in practice. Since the relationship between nodes in a graph is not necessarily constant, defining a simple discrete relationship between them (i.e. a discrete finite filter) is not easily achievable. Compounding this problem, the explicit spatial relationship between points is ambiguous, and depends on how a graph is embedded in space (which is not often provided as a parameter). As such, one typically turns to spectral methods or their approximations to compute the convolution of a "filter" with a graph [8, 23]. To this end, graph convolution is not necessarily compatible with the ideology presented by traditional convolutional methods; while compelling, it is difficult to compare deep non-Cartesian methods with graph learning approaches. While the geometry of non-Cartesian structures is different than that of the traditional tensor, it is still a *regular* structure; and as such, we seek to leverage pre-existing optimized implementations for operations on these domains.

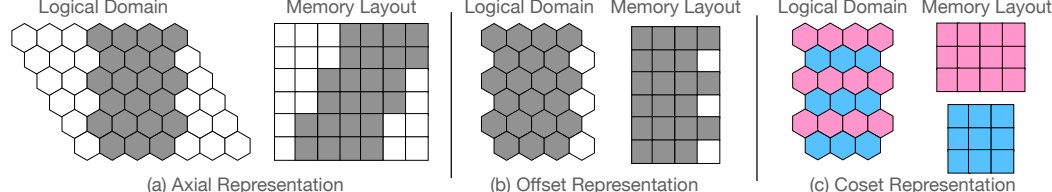

| Logical Domain | Memory Layout | Logical Domain | Memory Layout | Logical Domain | Memory Layout |
|---|---|---|---|---|---|

(a) Axial Representation    (b) Offset Representation    (c) Coset Representation

Figure 2: Three possible underlying representations for the memory of an image stored on a rectangular bounded region of the hexagonal lattice. The "Logical Domain" is the domain over which an operation operates, the shaded cells are the cells in which data is stored, and the white cells are empty (i.e. wasted space). The axial representation (a) treats the hexagonal domain as a transformed Cartesian domain, and requires a large amount of wasted space [34]. The offset representation (b) is more compact, but requires additional logic for the different rows of the image and may also waste space. Our coset representation decomposes the input lattice into a number of Cartesian lattices, is more compact than the offset representation in general, and works with arbitrary lattices.

## 3 Computing on Non-Cartesian Grids

In general, processing data on any non-Cartesian grid is simple, but technically cumbersome. While most algorithms are easy to state on a non-Cartesian discretization, the non-separable nature of these lattices makes it difficult to design efficient structures for storage and computation. We avoid this by simply ignoring the non-Cartesian structure; treating all processing as operations over separate Cartesian components. A similar approach is taken in some recent works in the non-Cartesian world [17, 16]. Figure 1 shows an example of this decomposition. The ideas in this paper are general, and we state them as $s$-dimensional theory. However, in practice, we implement this in 2D and 3D in NCDL.

**Lattices** We define a lattice as a subset of the $s$-dimensional integers $\mathbb{Z}^s$. Explicitly, we denote this with a full rank $s \times s$ integer matrix $\mathbf{L}$, and define a lattice as a set $\mathcal{L} = \mathbf{L}\mathbb{Z}^s \subseteq \mathbb{Z}^s$. Practically speaking, we are interested in a bounded subset of these integers. In the rest of this paper we assume this set is bounded within a rectangular region $\mathcal{R}$ of $s$-dimensional Euclidean space; we denote this bounded set as $\mathcal{L}_\mathcal{R}$. When $\mathbf{L} = \mathbf{I}$ then all the discussion reduces to the Cartesian lattice, and classical machine learning approaches trivially apply.

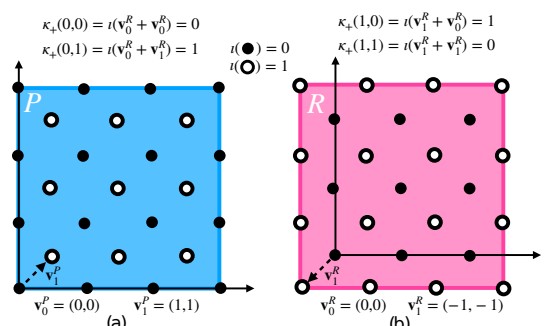

Figure 3: Two lattice tensors on the quincunx lattice. Each contains two tensors and two shifts (the first shift is always the trivial 0 shift). In general, on other lattices there may be more cosets and shifts. These two lattice tensors are *compatible*, that is, they share the same overall geometry, but are shifted.

The definition of a lattice usually invokes notions of combinatorial problems and theory. While we do require some lattice algorithms to facilitate our implementation, we instead mainly use lattices as data-structures. To each lattice site within $\mathcal{R}$, we assign a real (floating point) value; we also explicitly include a batch and channel index to facilitate common operations in deep learning. We combine these ideas in the notation

$$a_{b,c,\mathcal{L}_\mathcal{R}}[i, k, \mathbf{n}] \in \mathbb{R}, \ \forall \mathbf{n} \in \mathcal{L}_\mathcal{R}, 0 \le i < b, 0 \le c < k. \quad (1)$$

When the channel and batch indices are clear from context, we will write this as $a_{\mathcal{L}_\mathcal{R}}[i, k, \mathbf{n}]$, or $a[i, k, \mathbf{n}]$ when all parameters are apparent from context.

**The Coset Decomposition** The key observation that facilitates the mechanics of this paper is that any (integer) lattice structure can be written as a collection of Cartesian lattices. That is, there exists an integer $C$, an integer diagonal matrix $\mathbf{D} \in \mathbb{Z}^{s \times s}$ and integer shifts $\mathbf{v}_i, 0 \le i < C$ such that

$$\mathcal{L}_\mathcal{R} = \left( \bigcup_{i=0}^{C-1} \mathbf{v}_i + \mathbf{D}(\mathbb{Z}^+)^s \right) \cap \mathcal{R}. \quad (2)$$

This is the *Cartesian coset* decomposition. There are many possible coset decompositions; in general $\mathbf{D}$ need not be a diagonal matrix, however, this condition forces the cosets to be Cartesian. This particular representation allows us to store data on a lattice as multiple shifted traditional tensors (a.k.a. multi-dimensional arrays). To ensure that our algorithms maintain a consistent coset representation, we place a restriction on $\mathbf{v}_i$ and $\mathcal{R}$; we restrict all $\mathbf{v}_i \in \mathbf{D}[-1, 1]^s$. This puts a lower bound on the boundaries of the region $\mathcal{R}$, and helps avoid "coset creep" (in earlier iterations of this work, we noticed that coset vectors tend to uncontrollably walk around space if not somehow restricted).

**Lattice Tensor**   The *lattice tensor* is the simple, yet fundamental backbone of this work. A lattice tensor is a collection of tensors and integer vector shifts. We define this formally, as well as a small number of additional properties in the following:

**Definition 3.1** (Lattice Tensor). A *lattice tensor* consits of the bounded region $\mathcal{R}$, the collection of vectors $\mathbf{v}_i^{\mathcal{R}} \in \mathbf{D}[-1, 1]^s$, and the associated values on the lattice $a_{b,c,\mathcal{L}_{\mathcal{R}}}[l, k, \mathbf{n}]$. When obvious from context, we will simply write $\mathbf{v}_i$. We order the $\mathbf{v}_i$ lexically, according to their values $\mathbf{v} \mod \mathbf{D}$ (i.e. the vector in their equivalence class in $\mathbf{D}[0, 1]^s$). We denote the individual cosets of this lattice tensor as $\tilde{a}_j[l, k, \mathbf{m}]$, where $0 \le j < C$ is the coset index, and $\mathbf{m} \in (\mathbb{Z}^+)^s$.

The ordering of a lattice tensor's coset vectors is a small, but important part of the definition of a lattice tensor; it ensures that between two arbitrary rectangular regions $\mathcal{P}$ and $\mathcal{S}$ the set $\mathbf{v}_i^{\mathcal{P}} + \mathbf{D}\mathbb{Z}^s$ is always equivalent to $\mathbf{v}_i^{\mathcal{S}} + \mathbf{D}\mathbb{Z}^s$. When working with the coset representation it is important to know which coset lattice points belong to; this is formalized as follows:

**Definition 3.2** (Coset Index). Given some $\mathbf{n} \in \mathcal{L}_{\mathcal{R}}$, the coset index $\iota(\mathbf{n})$ is the integer such that $\mathbf{n} \in \mathbf{v}_{\iota(\mathbf{n})} + \mathbf{D}(\mathbb{Z}^+)^s$.

**Proposition 3.1** ($\kappa$-index). Given $i, j$ with $0 \le i, j < C$, for any $\mathbf{n} \in \mathcal{L}_{\mathcal{R}}, \mathbf{m} \in \mathcal{L}_{\mathcal{S}}$ with $\iota(\mathbf{n}) = i$ and $\iota(\mathbf{m}) = j$, it must be that both $\iota(\mathbf{n} - \mathbf{m})$ and $\iota(\mathbf{n} + \mathbf{m})$ are constant. We define $\kappa_{\pm}(i, j) := \iota(\mathbf{n} \pm \mathbf{m})$.

The intuition for the $\kappa$-index is simple, if we subtract or add lattice points on two cosets, the $\kappa$-index tells us the resulting coset index of the operation. Figure 3 shows an example of these concepts.

**Tensor Operations on Lattice Tensors**   For the most part, tensor arithmetic on lattice tensors is relatively straightforward. If two lattice tensors share the same $\mathbf{v}_i$ and $\mathcal{R}$, then arithmeteic can be performed directly on the underlying coset tensors. However, there may be cases in which we *should* be able to add two lattice tensors, but the differing coset vectors break the correspondence between two lattice tensors' coset. Figure 3 shows a simple case where this happens.

**Definition 3.3** (Compatibility). We say two lattice tensors $x_{b,c,\mathcal{L}_{\mathcal{R}}}$ and $y_{b',c',\mathcal{L}_{\mathcal{S}}'}$ are compatible if and only if $b = b', c = c', \mathcal{L} = \mathcal{L}'$ and $\exists\, \mathbf{k}_{\mathcal{R},\mathcal{S}} \in \mathbb{Z}^s$ such that $\mathcal{L}_{\mathcal{R}} = \{\mathbf{n} + \mathbf{k}_{\mathcal{R},\mathcal{S}} : \mathbf{n} \in \mathcal{L}_{\mathcal{S}}'\}$.

In practice, finding this correspondence is straightforward, we shift the set of coset vectors so that their centroid is the origin, then we check if all shifted coset vectors have a corresponding match. This allows us to define element-wise binary operations as operations over the underlying Cartesian tensors.

**Definition 3.4** (Binary Elementwise Operations). Given a binary operation $\odot$ defined over the elements of $\mathcal{R}$, two compatible lattice tensors $x_{b,c,\mathcal{L}_{\mathcal{R}}}$ and $y_{b,c,\mathcal{L}_{\mathcal{S}}}$, we define

$$(x_{b,c,\mathcal{L}_{\mathcal{R}}} \odot y_{b,c,\mathcal{L}_{\mathcal{S}}})[i, j, \mathbf{n}] := x_{b,c,\mathcal{L}_{\mathcal{R}}}[i, j, \mathbf{n}] \odot y_{b,c,\mathcal{L}_{\mathcal{S}}}[i, j, \mathbf{n} + \mathbf{k}_{\mathcal{R},\mathcal{S}}]. \tag{3}$$

Note that this always outputs a new lattice tensor on $\mathcal{L}_{\mathcal{R}}$.

**A Note About Reflection**   It is common to flip a tensor about an axis, either for data augmentation or for another common operation, like padding. However, an astute reader may notice that not all lattices have reflective symmetry.

**Definition 3.5.** A lattice $\mathcal{L}$ has reflective symmetry about the plane defined by $\mathbf{e}_d$ if and only if $\forall \mathbf{n} \in \mathcal{L}$ it is true that $-\mathbf{e}_d \cdot \mathbf{n} \in \mathcal{L}$ where $\mathbf{e}_d$ is the $d^{th}$ unit cardinal vector.

If a lattice tensor has reflective symmetry about an axis, materializing a reflected tensor is straightforward: flip the underlying tensor and correct the coset shift vectors.

**Lattice Tensor Padding**   Padding is a simple but fundamental operation in many data processing schemes. On the Cartesian lattice we simply extend the region of interest, then introduce the appropriate new elements. Padding a lattice tensor on $\mathcal{L}_\mathcal{R}$ is slightly more involved. Fix an axis $d$, and assume we wish to pad on the right by a unit of 1. In this case, we extend the support of $\mathcal{R}$ by the smallest amount on the right so as to introduce new lattice sites to the lattice tensor, we then pad the appropriate underlying tensors on the right to fill this new space. To pad on the left by a unit of 1, we again extend the support of $\mathcal{R}$ by the smallest amount on the left so as to introduce new lattice sites to the lattice tensor, any underlying tensors are padded to the left, and their corresponding coset vectors become $\mathbf{v}_i - \mathbf{e}_d$. If this moves a coset vector outside of the valid range $\mathbf{D}[-1, 1]^s$, then all vectors are uniformly shifted back within this range. We leave the details to our implementation, provided as supplementary material.

**Convolution**   The convolution operation naturally generalizes as an operation over lattices, we note this in the following definition.

**Definition 3.6.** For a lattice tensor $a_{\mathcal{L}_\mathcal{R}}$ and filter $f_{\mathcal{L}_\mathcal{P}}$ where the region $\mathcal{P}$ is strictly positive, we define convolution as

$$(a_{b,c,\mathcal{L}_\mathcal{R}} \star f_{c,k,\mathcal{L}_\mathcal{P}})[n, i, \mathbf{p}] := \sum_{j=0}^{c-1} \sum_{s \in \mathrm{supp}(f)} a[n, j, \mathbf{p} + \mathbf{s}] f[i, j, \mathbf{s}] \tag{4}$$

whose output lattice tensor is in $\mathcal{L}_\mathcal{S}$ where $\mathcal{L}_R = \mathcal{L}_\mathcal{S} \oplus \mathcal{L}_\mathcal{P}$ (here, $\oplus$ is the Minkowski-sum of two point-sets).

When $\mathcal{L}$ is the Cartesian lattice, then this reduces to the classical case. This definition excludes fused operations such as downsampling (strided convolutions) and any form of padding. We do this intentionally, as including these operations is quite complex, and supporting such fused operation would require a large additional implementation effort. Even as such, creating a dedicated implementation of this generalized convolution operation is a large challenge on its own [2]. Instead, we leverage pre-existing implementations on Cartesian lattices. We use the following proposition to achieve this.

**Proposition 3.2.** For a lattice tensor $a_{\mathcal{L}_\mathcal{R}}$ and filter $f_{\mathcal{L}_\mathcal{P}}$ where the region $\mathcal{P}$ is strictly positive, the result of the convolution $o_{b,k,\mathcal{L}_\mathcal{S}} := (a_{b,c,\mathcal{L}_\mathcal{R}} \star f_{c,k,\mathcal{L}_\mathcal{P}})$ can be written in terms of its output cosets as

$$\tilde{o}_i = \sum_{j=0}^{C-1} \tilde{a}_{\kappa_+(i,j)} \left[ \cdot + \delta(i,j) \right] * \tilde{f}_j \tag{5}$$

where $*$ is the traditional Cartesian convolution operator, and $\delta(i, j) := \mathbf{D}^{-1}(\mathbf{v}^\mathcal{R}_{\kappa_+(i,j)} - \mathbf{v}_i^\mathcal{R} - \mathbf{v}_j^\mathcal{S})$ is a constant by which we shift the appropriate coset of $a_{\mathcal{L}_\mathcal{R}}$.

A practical note about this operation is in order. We are typically limited to rectangular support for $\mathrm{supp}(\tilde{f}_j)$ in all popular implementations of Cartesian convolution. To address this, we simply take the "ZeroOut" approch [47], and zero out any inappropriate elements in the square filters' support.

**Pooling**   The pooling operator also naturally extends to lattice tensors, we note this in the following definition.

**Definition 3.7.** For a lattice tensor $a_{\mathcal{L}_\mathcal{R}}$ and filter geometry $\mathcal{L}_\mathcal{P}$ where the region $\mathcal{P}$ is strictly positive, lattice pooling is defined as

$$o_{b,c,\mathcal{L}_R}[l, m, \mathbf{n}] = \max_{\mathbf{s} \in \mathcal{L}_P} \{a[l, m, \mathbf{n} + \mathbf{s}]\} \tag{6}$$

whose output lattice tensor is in $\mathcal{L}_\mathcal{S}$ where $\mathcal{L}_\mathcal{R} = \mathcal{L}_\mathcal{S} \oplus \mathcal{L}_\mathcal{P}$.

We again, re-write this in terms of processing over Cartesian lattices in order to leverage existing frameworks for evaluation

**Proposition 3.3.** For a lattice tensor $a_{\mathcal{L}_\mathcal{R}}$ and stencil geometry $\mathcal{L}_\mathcal{P}$ where the region $\mathcal{P}$ is strictly positive, the result of the lattice pooling operation $o_{b,c,\mathcal{L}_\mathcal{R}}[l, m, \mathbf{n}] = \max_{\mathbf{s} \in \mathcal{L}_\mathcal{P}}\{a[l, m, \mathbf{n} + \mathbf{s}]\}$ can be written in terms of its output cosets as

$$\tilde{o}_i[l, m, \mathbf{n}] = \max_{0 \le j < C} \left\{ \max_{\mathbf{s} \in \mathbf{D}^{-1}(\mathcal{L}_\mathcal{P} - \mathbf{v}_j) \cap \mathbb{Z}^s} \left\{ \tilde{a}_{\kappa_+(i,j)} \left[\mathbf{n} + \mathbf{s} + \delta(i,j)\right] \right\} \right\} \tag{7}$$

where the maximum operation running over the set $\mathbf{D}^{-1}(\mathcal{L}_\mathcal{P} - \mathbf{v}_j) \cap \mathbb{Z}^s$ is the traditional max pool operator, restricted to the $j^{\text{th}}$ coset of the stencil.

**Downsampling** Downsampling (i.e. discarding data then reducing resolution) is another basic operation that we generalize. The most general case for downsampling is to systematically remove samples in such a way that the lattice property is preserved. We take the standard path to define down/up sampling about an integer matrix $\mathbf{S}$ with $0 < \det \mathbf{S} \in \mathbb{Z}$. We obtain the sub-sampled lattice by restricting to the new set $\mathcal{Q} := \mathbf{S}\mathcal{L}$. Loosely, for the new lattice $\mathcal{Q}$ we obtain the coset decomposition, then subsample the input lattice tensor according to the new decomposition. This yields a new lattice tensor.

**Upsampling** Upsampling introduces new datapoints (initialized to zero) at a higher resolution lattice. We denote this with a matrix $\mathbf{S}^{-1}\mathbf{L} \in \mathbb{Z}^{s \times s}$. This employs the same mechanics as downsampling, but rather than removing points, we introduce new points according to the lattice in question. The mechanics of this are similar to downsampling, and we leave the details to the supplementary implementation.

### 3.1 Derivative Computation

Since gradient computation is an important feature that must be supported by any deep learning framework, it is crucial to discuss gradient computation from the perspective of lattice tensors. For a given loss function $h$, and the lattice tensor $a$ the gradient $\partial h / \partial a$ is simply the lattice tensor consisting of the gradients of the individual cosets, that is, $\partial h / \partial \tilde{a}$. Computing the gradient of any element-wise operation is straightforward; we simply operate independently on the coset tensors. However, for the remainder of the operations discussed so far, we require more care. We start with convolution.

**Proposition 3.4.** For a lattice tensor $a_{\mathcal{L}_{\mathcal{R}}}$ and filter $f_{\mathcal{L}_{\mathcal{P}}}$ where the region $\mathcal{P}$ is strictly positive, with convolution defined as

$$o[n, i, \mathbf{p}] := (a_{b,c,\mathcal{L}_{\mathcal{R}}} \star f_{c,k,\mathcal{L}_{\mathcal{P}}})[n, i, \mathbf{p}] \tag{8}$$

whose output lattice tensor is in $\mathcal{L}_{\mathcal{S}}$ where $\mathcal{L}_R = \mathcal{L}_{\mathcal{S}} \oplus \mathcal{L}_{\mathcal{P}}$, the derivatives of the loss $h$ with respect to the filter and input lattice tensor are given by

$$\frac{\partial h}{\partial a[n, i, \mathbf{k}]} = (o \star \overline{f_{c,k,\mathcal{L}_{\mathcal{P}}}})[n, i, \mathbf{k}], \tag{9}$$

$$\frac{\partial h}{\partial f[i, j, \mathbf{k}]} = \sum_{n=0}^{b-1} \left( \sum_{\mathbf{p} \in \mathcal{L}_{\mathcal{S}}} \frac{\partial h}{\partial o[n, i, \mathbf{p}]} \cdot a[n, j, \mathbf{p} + \mathbf{k}] \right), \tag{10}$$

where $\overline{f}$ mirrors the filter and swaps the channels, i.e. $\overline{f}[i, j, \mathbf{k}] := f[j, i, -\mathbf{k}]$.

Note that it is common to state the filter gradient as a convolution. We avoid introducing new notation to describe this fact and simply note this here.

**Proposition 3.5.** For a lattice tensor $a_{\mathcal{L}_{\mathcal{R}}}$ and filter geometry $\mathcal{L}_{\mathcal{P}}$ where the region $\mathcal{P}$ is strictly positive, lattice pooling

$$o_{b,c,\mathcal{L}_R}[n, i, \mathbf{k}] = \max_{\mathbf{s} \in \mathcal{L}_P} \{a[n, i, \mathbf{k} + \mathbf{s}]\} \tag{11}$$

whose output lattice tensor is in $\mathcal{L}_{\mathcal{S}}$, has the gradients given by

$$\frac{\partial h}{\partial a[n, i, \mathbf{k}]} = \sum_{\mathbf{p} \in \mathcal{L}_{\mathcal{P}}} \frac{\partial h}{\partial o[n, i, \mathbf{k} - \mathbf{p}]} \cdot \mu[n, i, \mathbf{k} - \mathbf{p}, \mathbf{p}] \tag{12}$$

where

$$\mu[n, i, \mathbf{p}, \mathbf{q}] := \begin{cases} 1 & \text{if } \max_{\mathbf{s} \in \mathcal{L}_P} \{a[n, i, \mathbf{p} + \mathbf{s}]\} = a[n, i, \mathbf{p} + \mathbf{q}] \\ 0 & \text{otherwise.} \end{cases} \tag{13}$$

The proofs of these facts are direct consequences of the chain rule, and can be found in the Appendix.

**Derivatives of Padding, Downsampling and Upsampling** The loss gradient of the downsampling operation is upsampling (taking care to respect the original dimensions of the original lattice tensor) and vice versa. The gradient of the padding operation simply truncates the padded gradient tensor, discarding any gradient signal outside of the original geometry of the tensor from the forward pass. For the most part, we leverage the automatic differentiablity of PyTorch to compute derivatives of these operations [31]. However, more streamlined implementations using the forms above may save memory in future iterations of this work, since less temporary tensors must be saved during the forward pass.

# 4 Exploring non-Cartesian Networks

In this section, we delineate the experiments we conducted to evaluate NCDL. First, we compare NCDL with the closest existing hexagonal convolution library [34]. Subsequently, we explore the use of non-dyadic down/up sampling within bottlenecked architectures. To the best of our knowledge, this operation has not been explored in machine learning. All experiments are conducted on an AMD Ryzen 9 3900X (3.8GHz) with 128GB of DDR4 RAM operating at 3200 MHz, accompanied by an NVIDIA RTX 3090 with 24GB of RAM.

## 4.1 Hexagonal Convolution

While NCDL focuses primarily on integer lattices, the hexagonal lattice exhibits the same coset structure as the quincunx lattice. This permits the storage of a hexagonal image on a quincunx lattice — convolution can be implemented via the selection of a suitable filter on the quincux lattice. We assess the efficacy of this method in comparison to HexagDLy [34], which, among all other related works, is conceptually most similar to our approach.

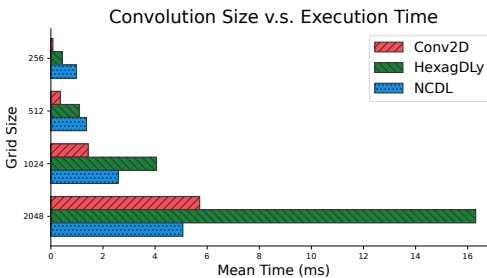

Figure 4: Average run time in milliseconds (ms) as grid size increases, averaged over 100 runs, bench-marked on GPU. Variance is negligible.

**Evaluation** Figure 4 illustrates the performance trend as the grid size increases. Here, all grids are "equivalent" to the Cartesian grid. That is, we choose our hexagonal domain/lattice tensors to consume roughly the same amount of memory as the Cartesian grid. For example, if our Cartesian grid is $256 \times 256$, then each coset of our lattice tensor has a dimension of $182 \times 182$. We fixed input and output channels at 32 and 64 respectively (simulating an early stage in a convolutional pipeline). We limit our experiment to the case where the convolution kernel is a hexagonal 1-ring/1-neighborhood filter. We set the stride to 1, as NCDL does not support fused strides with convolution. It is important to note that we employ the same filter sizes as HexagDLy, but our convolutions occur over the separate cosets of the lattice, which are smaller than the grids over which HexagDLy operates.

There are a few things to note about this experiment. For smaller grid sizes, NCDL exhibited slower performance compared to HexagDLy. This is likely due to the inherent overhead of managing the coset structure and the overhead of GPU kernel queueing. However, beyond a threshold, NCDL vastly outperforms HexagDLy. This is likely due to the fact that HexagDLy operates over larger grids (even though it employs strided convolutions). This effectively increases the computational load, particularly in processing larger inputs. While the strides in HexagDLy's implementation help in reducing the spatial dimensions of the output, the fundamental operation over larger grids does demand additional computational resources. Interestingly, only in one scenario (2048) did a non-Cartesian implementation outperform the Cartesian 1-ring filter. This finding is particularly significant, considering that the Cartesian convolution employs a larger filter size and is thus expected to involve more computation.

## 4.2 Non-dyadic Downsampling and the Quincunx Lattice

It is somewhat expected that convolution on hexagonal images yields superior results in aspects such as speed or accuracy. Nonetheless, non-Cartesian lattices may enhance performance without necessitating a change in approximation space through other means. In this second set of experiments, we investigate the application of non-dyadic downsampling. That is, as data flows through the network, it starts as a tensor on a Cartesian lattice; subsequent layers perform a convolution, then a downsample operation according to the following sub-sampling matrix $\mathbf{S} = [-1\ 1; 1\ 1]$. This generates a lattice tensor on the Quincunx lattice. A subsequent convolution (followed by a downsample operation) results in a lattice tensor on the Cartesian lattice. Different levels of convolution alternate between these two grids (analogous to [19]), offering a more gradual reduction in resolution.

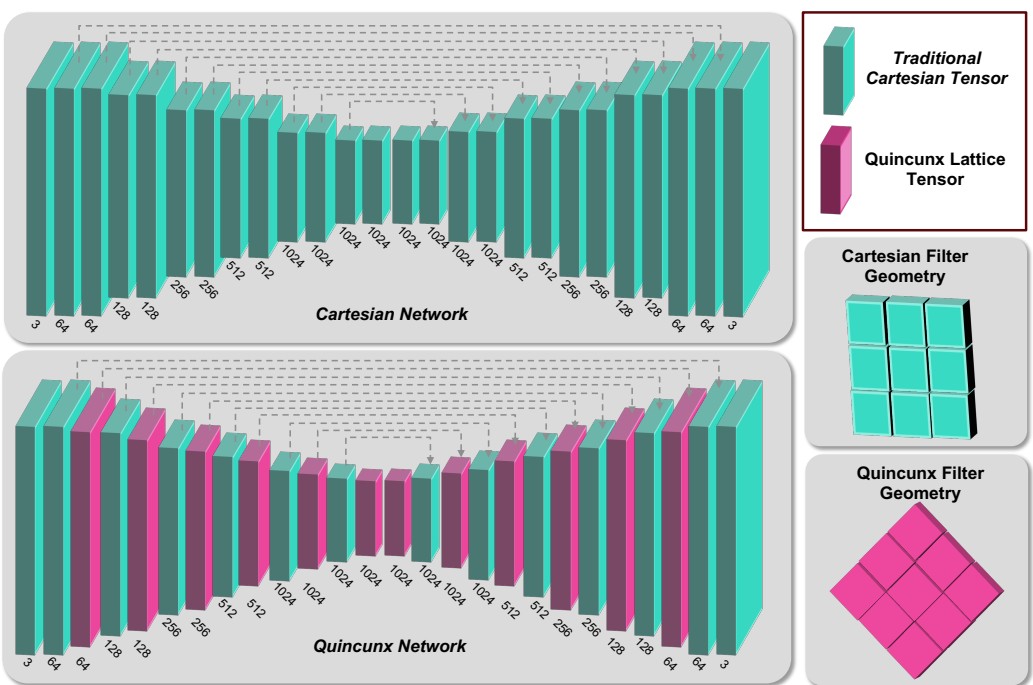

Figure 5: The network structures we propose. The auto-encoder experiments we conduct omit the skip connections between the layers. To maintain parity with the quincunx design, additional skip connections were added to the Cartesian case. Both networks share the same amounts of parameters.

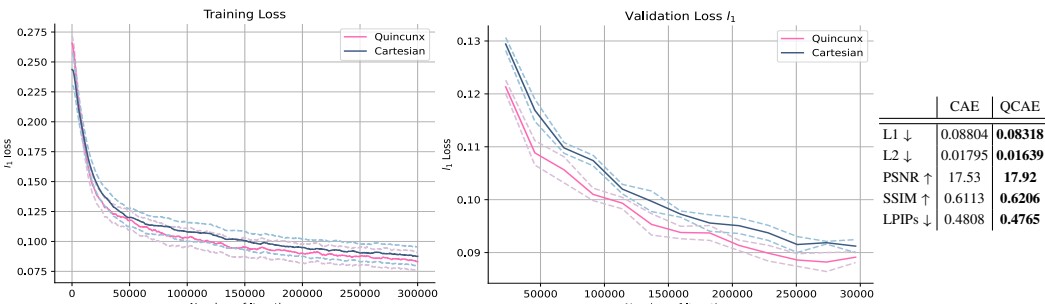

|  | CAE | QCAE |
|---|---|---|
| L1 ↓ | 0.08804 | **0.08318** |
| L2 ↓ | 0.01795 | **0.01639** |
| PSNR ↑ | 17.53 | **17.92** |
| SSIM ↑ | 0.6113 | **0.6206** |
| LPIPs ↓ | 0.4808 | **0.4765** |

Figure 6: Error plots for training and validation. The QCAE marginally outperforms in training, and outperforms with a wider margin in the validation. The table on the right shows the final model performance with respect to various error metrics; superior performance is noted in bold-font.

**Quincunx Auto-encoder**    In this experiment, we establish a rudimentary autoencoder structure that progressively downsamples to a latent space before upsampling back to the image space. We first establish a baseline experiment, incorporating two Cartesian convolutions with $3 \times 3$ filters, followed by a downsample (we maintain two convolutions to preserve parameter parity with the quincunx case). The Quincunx Convolutional Auto-Encoder (QCAE) lattice encompasses one convolution, a downsample onto the Quincunx lattice, one convolution, and a downsample onto the Cartesian lattice. Figure 5 shows the exact structures we design.

We train our models using the CelebA dataset [30]. Employing a straightforward L1 loss, we measure validation L1, L2, PSNR, SSIM [41] and LPIPs metrics [46]. The network is trained with the Adam optimizer, default parameters, and a batch size of 8. We train for 300,000 iterations, as convergence was observed at this point, and take an average of 5 runs. This represents a total of approximately 160 hours of training time for all experiments.

**Salient Object Detection with a Quincunx U-Net**    In this experiment, we extend the rudimentary auto-encoder from the previous experiment by simply adding skip connections. Figure 5 shows the exact structures we design, and the dotted lines show the additional skip connections that we

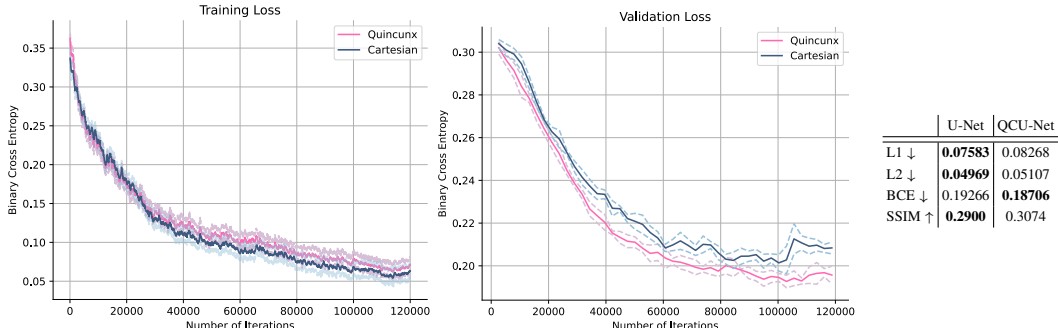

Figure 7: Error plots for training and validation. The QCU-Net marginally fits to the data in the training set worse than the strictly Cartesian U-Net. However, the QCU-Net outperforms in generalization for this task. The table on the right shows the validation error at the end of training. Better performance is noted in bold-font.

introduce for this example. We train our models on the DUTS salient object detection dataset [40], and employ a straightforward binary cross-entropy (BCE) loss; we measure validation BCE, L1, L2 and SSIM. The network is trained with the Adam optimizer, with default parameters, and a batch size of 8. We train for 120,000 iterations, as convergence was observed in the validation data at this point.

**Discussion**    In our experiments, introducing a quincunx downsampling shows a noteable difference across our tasks. For the autoencoder experiment, we note a small improvement in performance for no additional cost in parameters (Figure 6); replacing a traditional Cartesian convolution with a downsample operation followed by a Quincunx convolution should also be more computationally efficient, since certain convolutions occur on smaller grids. However, due to implementation overhead, we note that this is currently not true. The U-Net experiment is interesting in that it does not show distinctly better performance in training, but validation performance improves for the BCE metric (Figure 7). It is somewhat surprising that other metrics do not reflect the improved validation entropy; however minimizng BCE does not strictly imply that other pixel-wise metrics will improve, due to the way in which BCE weights pixelwise classification errors.

There are a few potential reasons for the improved quantitative performance of these networks. The downsampling scheme is more gradual in a QCAE. This likely leads to a more gradual reduction of information as it passes through the bottleneck of the network. Important information is less likely to be "missed" by this more gradual reduction. The second, more simple explanation may be that the aliasing introduced by the quincunx downsampling operation is much less strong, thereby providing a stronger signal as data pass through lower resolution layers of the network; aliasing is a well-known enemy of neural networks [45].

## 5   Future Work and Conclusions

This work has presented a general approach to enable the use of non-Cartesian lattices in machine learning. Doing so has opened a new avenue of research by introducing a new twist on a fundamental concept in machine learning. There are many possible avenues for future work: we plan on exploring this in the context of models at scale, for example in generative approaches like GANs or diffusion models [5, 20]. Additionally, modern components, such as the attention mechanism [39], must be generalized to this new structure. There is also still a good amount of performance left on the table. Decomposing convolution across the different cosets separates the convolution operation into disjoint operations that could, for example, take place on completely separate hardware. These separate operations could be scheduled across different memory chips, for example, to avoid bank conflicts or other complications that arise in GPU memory controllers.

## Acknowledgements

We would like to thank Richard Zhang and Derek Nowrouzezahrai for their involvement in earlier less successful iterations of this work and for pushing us to finish this work.

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
