# OpenReview forum: "NCDL:  A Framework for Deep Learning on non-Cartesian Lattices"
_NeurIPS.cc/2023/Conference — NeurIPS 2023 poster_

### Official Review · Reviewer_Y55j · 2023-07-01

**Soundness:** 4 excellent
**Presentation:** 3 good
**Contribution:** 4 excellent
**Rating:** 8
**Confidence:** 4

**Summary:**

This paper generalizes common machine learning operations from Cartesian lattices to other regular lattices, such as the hexagonal lattice. The authors argue that the Cartesian lattice is a sub-optimal representation for important natural signals, and that operating on their non-Cartesian structure natively leads to more efficient implementations and better results.

The authors further promise to release a software library for non-Cartesian deep learning, and include an experimental section with implementation details and efficiency arguments, as well as experimental results on various computer vision tasks.

**Strengths:**

The contribution is very novel in the sense that the field of deep learning on lattices is under-explored. The author's work is likely to have a big impact on this area of machine learning, as providing a complete optimized library with non-Cartesian version of common lattice operations with speed-up future research significantly.

**Weaknesses:**

The paper is short, with the authors being allowed one more page of content. They could improve the discussion by expanding, for example, on the computation of the derivatives.

Are the backward passes of all operations naturally handled by PyTorch or did it require manual implementations? What about numerical stability?

**Questions:**

The area of non-Cartesian deep learning could reasonably be considered a sub-field of geometric deep learning, which also includes deep learning on graphs, and models that leverage group symmetry. The authors should consider comparing lattice neural network approaches with GNNs (similar to experiments using sub-pixel graphs), as the graph models should be able to handle the non-Cartesian grid (albeit differently and with less inductive bias).

**Limitations:**

Some aspects are lacking, as explained above:
- Derivatives and numerical stability
- Comparison with other non-Cartesian models such as GNNs and group equivariant neural networks

---

> ### Author Rebuttal · Authors · 2023-08-08
>
> Thank you for your review : ). Please, see the global rebuttal, it should address most your comments.
>
> I believe the only issue left unaddressed is the question about numerical stability. This is a good catch, and something we didn't touch on in the paper. Since we leverage PyTorch for (almost all) of our operations, we inherit all the properties of PyTorch's underlying implementations. Lattice tensor convolution, for example, is the sum of a small number number of cartesian convolutions. If we assume PyTorch's implementation of convolution is stable, then we can reasonably assume that the resulting lattice convolution is stable. This is not a completely formal argument, but could be formalized better if appropriate for the final submission.

---

> > ### Comment · Reviewer_Y55j · 2023-08-20
> >
> > Thank you for your reply.
> >
> > Indeed providing the theory for the computation of the derivatives is of high interest to the community for:
> > - intellectual reasons
> > - efficiency as mentioned in the rebuttal (fusing operations etc)
> > - checking numerical stability and/or implementing alternative direct computations to improve stability or handle edge cases

---

### Official Review · Reviewer_Pd5c · 2023-07-07

**Soundness:** 3 good
**Presentation:** 3 good
**Contribution:** 3 good
**Rating:** 5
**Confidence:** 3

**Summary:**

This works introduces a framework as well as software for computing convolutions on non-Cartesian lattices. The method is compared to existing software for hexagonal lattices as well as on image data.

**Strengths:**

The method is put in a strong theoretical framework that also explores the very important up and down sampling operations on non-uniform grids. The authors also make available open-source software that is more general and whose performance seems much better than anything available.

**Weaknesses:**

In the context of scientific machine learning, there have been similar ideas explored on how to make architectures which work on arbitrary grids, for example, https://arxiv.org/abs/2207.05209, https://arxiv.org/abs/2305.19663, and I am sure there are others. Furthermore, graph neural networks can also perform convolutions or arbitrary grids, for example, https://arxiv.org/abs/1704.01212, among many, many others. It would be good to mention some of these and even include some comparisons.

**Questions:**

The motivation for images on hexagonal grids are is a bit unclear to me. I always think of images as living on Cartesian grids, so it is somewhat strange to consider them on hexagonal grids instead. Why is this useful? Furthermore, where do you see most of the applications for this? Would be great to include possible large scale applications that lead a vision for future work.

**Limitations:**

The authors have adequately addressed limitations and potential negative societal impact.

---

> ### Author Rebuttal · Authors · 2023-08-08
>
> Thank you for your review : ). With respect to the missing references, see the main rebuttal.
>
> This work does indeed require you to bend the notion that images are composed of square pixels. Given an isotropically band-limited image/function (which is really most natural images) it is possible represent that image with ~14% less samples on a hexagonal grid compared to a Cartesian grid [1]. This is perhaps surprising from the perspective of computer science, but when you look to nature, it is less surprising. Many structures that appear in nature are hexagonal; the photo-receptor cells in your retina are arranged in a nearly hexagonal pattern. There's a relatively large body of work on this, it is worth taking a look at both [1] and [2] if you are interested (also, some of the references in our background section expand on this higher dimensions).
>
> It's also worth noting that NCDL is not limited to the Hexagonal grid. We support any regular (integer) lattice structure. This is notable because it allows us to start with data on a Cartesian lattice, then move to another lattice (quincunx, for example, see the experiment at the end of the current submission). This provides a much smoother transition to lower resolution representations.

---

### Official Review · Reviewer_uhBv · 2023-07-09

**Soundness:** 4 excellent
**Presentation:** 4 excellent
**Contribution:** 4 excellent
**Rating:** 8
**Confidence:** 4

**Summary:**

This paper introduces a high-quality software extension for PyTorch that enables seamless computations with non-Cartesian lattices for 1D, 2D and 3D images. The key observation made by the authors of this work is that **non-Cartesian lattices can be decomposed as "sums" of Cartesian lattices**: up to some clever refactoring (which makes up the core numerical code of the proposed software package), efficient computations on non-Cartesian lattices can directly leverage the standard (and highly optimized) PyTorch/cuDNN implementations of e.g. convolutional layers.

This implies that **the proposed software package is both efficient and easy to maintain in the long run**.

**Strengths:**

- The authors tackle an interesting and **very original topic**. Non-Cartesian lattices are indeed fundamental to low-level image processing but essentially absent from the machine learning literature.

- The paper is **extremely well written**, with clear figures, attention paid to details and a satisfying evaluation. I haven't tried to run the code provided in the supplementary materials (too many papers to review at once!), but this is **clearly a high-quality software package** with a clean structure, a full test suite and extensive documentation.

- This package targets successfully one specific and interesting operation in image processing, providing a **useful extension to our common toolbox via a neat and clever software package**. This is more than what most papers (never mind submissions) can provide and, in my opinion, clearly warrants publication at NeurIPS.


**Weaknesses:**

Reviewers can always ask for more (outstanding deep learning experiments, better run times, support for all types of attention layers and niche hardware architectures, etc.)... But realistically, I am very happy with the paper as submitted.

**Questions:**



**Limitations:**

---

> ### Author Rebuttal · Authors · 2023-08-08
>
> Thank you, I believe no comments are necessary from me, here.

---

> > ### Comment · Reviewer_uhBv · 2023-08-10
> > **Acknowledgement of the rebuttal**
> >
> > You are welcome. I am very satisfied with the paper: having read all reviews and rebuttals, I am convinced that this submission is a very clear accept. Good luck!

---

### Official Review · Reviewer_aoAr · 2023-07-10

**Soundness:** 2 fair
**Presentation:** 2 fair
**Contribution:** 2 fair
**Rating:** 4
**Confidence:** 3

**Summary:**

he authors claim that the concept of tensors, a fundamental cornerstone in machine learning, assumes data are organized on Cartesian grids. They further suggest that alternative non-Cartesian representations may be more beneficial in certain situations. One case is when the data is inherently non-Cartesian — for example, the raw R/G/B image data from most imaging sensors are follows a quincunx (checkerboard) structure in the green channel. Another case is when non-Cartesian grids show superior performance in certain aspects — for example, the hexagonal lattice is known as the optimal sampling lattice in 2D.
The authors hereby propose a framework and a software library that introduces standard machine learning operations on non-Cartesian data. The new data structure is called lattice tensor and the software library is called Non-Cartesian Deep Learning (NCDL).


**Strengths:**

1. In the introduction section, the authors provided sufficient context on why non-Cartesian grid structures may be superior to Cartesian counterparts under certain circumstances.
2. We shall appreciate the effort in designing the memory-efficient coset representation for operations on the hexagonal lattice (shown in Figure 2).


**Weaknesses:**

While custom tensor definitions and operations for non-Cartesian data representation are theoretically pleasing, the authors have not shown clearly the potential applications and impact. It is not obvious which datasets and/or standard machine learning tasks will directly benefit from the non-Cartesian representation. A table detailing some typical use cases will be utterly helpful.
The experiments/comparisons performed are not convincing enough. The two main results shown in the submission are (1) runtime of convolution operation at different grid sizes, and (2) loss curves of a Cartesian vs. Quincunx auto-encoder on CelebA (celebrity faces) dataset.

In the first experiment, while the authors compare the runtime against the standard Cartesian Conv2D and another non-Cartesian baseline (HexagDLy), they have only investigated the convolution operation but skipped the other operations such as pooling, downsampling, upsampling, gradient computation, and back-propagation. Further, no comment has been made on numerical correctness or precision.
In the second experiment, the authors aim to show superior performance of Quincunx auto-encoder for image reconstruction. It is a bit weak as the experiment is only performed on one task over one dataset. It may be helpful to include a few more datasets — they don’t even need to be big one, e.g., STL-10, SVHN, LSUN will be sufficient. Besides, the only metrics shown are L1 and SSIM on the validation set. I would recommend including other metrics such as L2, PSNR, FID, and perceptual distance.


**Questions:**

1. Continuing on Weakness #1, it will be very helpful if the authors can construct a table which outlines several typical use cases, with the following information: (1) domain (vision/language/graph/etc), (2) dataset name and description of data format, (3) machine learning task (classification/segmentation/reconstruction/retrieval/etc), (4) preferable non-Cartesian representation, (5) short explanation on why it’s better than Cartesian. Do you think it is a reasonable idea, or is there a better way to demonstrate the applications and impact?

2. Open-ended question: will it be better to embed the proposed method in hardware?


**Limitations:**

While it has been covered in previous sections, the main limitations are:
-Unclear potential application and impact.
-Insufficient of experiments and comparisons.

---

> ### Author Rebuttal · Authors · 2023-08-08
>
> ###  "It will be very helpful if the authors can construct a table which outlines several typical use cases, with the following information: (1) domain (vision/language/graph/etc), (2) dataset name and description of data format, (3) machine learning task (classification/segmentation/reconstruction/retrieval/etc), (4) preferable non-Cartesian representation, (5) short explanation on why it’s better than Cartesian. Do you think it is a reasonable idea, or is there a better way to demonstrate the applications and impact?"
>
> This is a good suggestion, but difficult to execute because some of these points venture into research territory. I can briefly comment on how NCDL can affect each of these aspects.
>
>  * **domain**) The applicable domains here are any vision problem or any problem involving volumetric data. One key observation that bares repeating is that your data do not need to reside on a Cartesian grid to use these approaches. The non-dyadic downsampling operation takes data from one lattice configuration and places it in another (for example, Cartesian to Quincunx). Depending on the source and target lattice, this operation discards much less data compared to a standard stride=2 convolution. This would be a relatively large list of problems.
>
>  *  **datasets and data formats**) My assumption is that you would want to see how a non-Cartesian approach would be more appropriate with a given dataset? There are cases where data may be structured inappropriately for a specific grid. For example, if the data in question is pixel art, and we attempt to represent that pixel art in a hexagonal domain, this will clearly be beaten by a Cartesian approach. However, there may be some other network architecture that both 1) includes non cartesian convolution, and 2) outperforms a purely Cartesian approach. Again, this is something we are actively investigating as future research.
>
> * **non-Cartesian representation**. The lattice tensor is always our preferred representation. There is no compelling reason to use a different data-structure that can support arbitrary lattices as the lattice tensor does. Even if we limit ourselves to strictly hexagonal lattices, then there is no data-structure that has the flexibility needed for appropriately padding data before convolution (or pooling, or any other operation that consumes part of the data's spatial extent)
>
>  * **reason for using a non-Cartesian approach**. In the context approximation theory there are good arguments as to why non-Cartesian approaches should be superior to Cartesian approaches. In the context of machine learning, I'm not sure if these arguments hold, this needs to be assessed on a problem-by-problem basis.  NCDL introduces a new primitive that adds another degree of freedom to network architecture design, how that degree of freedom will affect results is not 100% clear. To us, this is a very attractive avenue for future work.
>
> We can add a table detailing some of what you want. This is completely doable and reasonable, however it will be based on speculation, and likely belongs in the future work section.
>
> ### "Will it be better to embed the proposed method in hardware?"
> We are confident that there would be a benefit to specialized hardware, yes. How to do this is a subject of future research. For example, there are necessarily parallel computations that can be exploited (for convolution, there are completely independent convolutions happening for one given operation). Without giving too much away, I can say that splitting grids onto physically separate memory chips could allow for more efficient use of the bandwidth of those chips. This is something we are actively thinking about.
>
> ### "Insufficient of experiments and comparisons."
> We will expand our set of test metrics, and add another application/dataset to compare with.
>
> ### "Unclear potential application and impact."
> We do not agree with this point. This work adds a set of new primitives that are applicable to many problems that use convolution. However, we do agree that more evaluation would help drive this point home.

---

### Author Rebuttal · Authors · 2023-08-08

First of all, I’d like to thank the reviewers for their time and detailed reviews. We will first address comments common to multiple reviewers.

###  Derivatives
Multiple reviewers pointed out some variation of concern towards the derivative computation. To clarify this, all of the computation for derivatives are handled by PyTorch. In earlier iterations of this work, some derivative computation was explicit, however we opted to rely on PyTorch for derivative computation; writing a new layer with a custom backwards pass (especially for something like the LatticeConvolution layer), is non-trivial (we noticed no significant performance difference when doing this for some test cases).

There would, however, be a benefit in terms of memory consumption, as fewer auxiliary tensors need to be kept for the backwards pass. We will expand the discussion around derivative computation, then provide the additional theory for these computations (at least) in the supplementary material.

### Graph convolution and other missed references:
We missed this connection. We will provide, at the very least, a discussion on graph convolutions and how they fit in with our work. We will cite the relevant information in the following works
* https://arxiv.org/abs/2207.05209
* https://arxiv.org/abs/2305.19663
* https://arxiv.org/abs/1704.01212

### References:
 [1] The processing of hexagonally sampled two-dimensional signals, Proceedings of the IEEE 67 (6) (1979) 930–949. doi:10.1109/PROC.1979.11356.

 [2] Middleton, Lee, and Jayanthi Sivaswamy. Hexagonal image processing: A practical approach. Springer Science & Business Media, 2005.

---

### Decision · Program_Chairs · 2023-09-21

**Decision:**

Accept (poster)

**Comment:**

The authors propose a framework and a software library that introduces standard machine learning operations (i.e. convolutions) for non-Cartesian data. The majority of reviewers were very positive about this work. While there was some concern about potential applications, this work adds a set of new primitives that could be applied to many problems with non-Cartesian grid structures.